# Mucosal application of the broadly neutralizing antibody 10-1074 protects macaques from cell-associated SHIV vaginal exposure

Karunasinee Suphaphiphat [1], Delphine Desjardins [1], Valérie Lorin [2], Nastasia Dimant[1], Kawthar Bouchemal[3], Laetitia Bossevot [1], Maxence Galpin-Lebreau[1], Nathalie Dereuddre-Bosquet [1], Hugo Mouquet [2], Roger Le Grand [1] & Mariangela Cavarelli [1]✉

Passive immunization using broadly neutralizing antibodies (bNAbs) is investigated in clinical settings to inhibit HIV-1 acquisition due to the lack of a preventive vaccine. However, bNAbs efficacy against highly infectious cell-associated virus transmission has been overlooked. HIV-1 transmission mediated by infected cells present in body fluids likely dominates infection and aids the virus in evading antibody-based immunity. Here, we show that the anti-N-glycans/V3 loop HIV-1 bNAb 10-1074 formulated for topical vaginal application in a microbicide gel provides significant protection against repeated cell-associated SHIV$_{162P3}$ vaginal challenge in non-human primates. The treated group has a significantly lower infection rate than the control group, with 5 out of 6 animals fully protected from the acquisition of infection. The findings suggest that mucosal delivery of potent bnAbs may be a promising approach for preventing transmission mediated by infected cells and support the use of anti-HIV-antibody-based strategies as potential microbicides in human clinical trials.

HIV-1 is present in the semen and other infectious body fluids, such as blood, vaginal secretions, and breast milk, in two forms: cell-free virions and infected cells (the so-called cell-associated virus)[1–3]. Both forms play a role in viral transmission and dissemination, although the latter has been overlooked in studies evaluating HIV-1 transmission and related prevention measures. In vivo macaque studies have shown that cell-associated virus is the primary means of vaginal and colorectal transmission of SIV[4,5] and at least a proportion of new HIV-1 infections in humans are initiated by infected cells[6,7]. Prevention strategies targeting both mechanisms of transmission are desirable to effectively control viral spread. However, cell-associated transmission represents

a major challenge in the fight against HIV-1 as it is considered a mechanism through which the virus, protected within the infected cells, can evade antibody-mediated neutralization[8–14].

Broadly neutralizing antibodies (bNAbs) against HIV-1 transmission have garnered significant attention in recent years as promising molecules for therapeutic or prophylactic interventions and are considered an ideal property of an HIV-1 vaccine. These antibodies can target conserved epitopes on the virus, thereby potentially neutralizing a broad range of HIV-1 strains[15]. In addition, bNAbs exert their protective activity through Fc-dependent functions, including antibody-dependent cellular cytotoxicity and activation of the complement[16–19].

[1]Université Paris-Saclay, Inserm, CEA, Center for Immunology of Viral, Auto-immune, Hematological and Bacterial diseases (IMVA-HB/IDMIT), Fontenay-aux-Roses & Le Kremlin-Bicêtre, France. [2]Laboratory of Humoral Immunology, Institut Pasteur, Université Paris Cité, INSERM U1222, 75015 Paris, France. [3]Chimie ParisTech, PSL University, CNRS, Institut de Recherche de Chimie Paris, CNRS UMR 8247, 75005 Paris, France. ✉e-mail: mariangela.cavarelli@cea.fr

In both non-human primates (NHPs) and humanized mice, *cell-free* HIV-1 infection can be prevented when animals are given either topical or systemic immunoprophylaxis[20–29], even after repeated challenges[30,31]. Additionally, bNAbs have been shown to significantly reduce the frequency of viral rebound after interruption of antiretroviral therapy[32,33], indicating a potential role in the maintenance of viral suppression. Several clinical trials have also been conducted to evaluate the safety and efficacy of bNAbs in human populations[34,35]. These trials have employed different dosing regimens, routes of administration, and bNAb candidates, and have provided valuable information on the feasibility of bNAbs as a treatment option. Some trials have demonstrated that potent bNAbs can effectively reduce viral loads and provide durable antiviral activity, while others have shown that the delivery of bNAbs may be limited by the development of viral resistance and the requirement for repeated administration[36–39].

To date, only one study has evaluated the efficacy of the PGT121 bNAb against cell-associated transmission of the simian-human immunodeficiency virus (SHIV) in macaques[40]. The Ab and the infected cells were both administered intravenously and the study demonstrated partial protection, possibly due to insufficient initial concentrations of PGT121 or Ab waning. A follow-up study demonstrated that Fc-mediated functions of PGT121 were not necessary for the conferral of protection against cell-associated SHIV[41].

Even though the majority of HIV-1 sexual transmissions occur through the vaginal or rectal mucosa[42], as yet, there is no indication that bNAbs can prevent in vivo mucosal transmission mediated by infected cells. The 10-1074 bNAb targeting a carbohydrate-dependent epitope in the V3 loop of the HIV-1 envelope spike significantly inhibited syncytia formation between uninfected and infected T cells and the transfer of viral material through the virological synapse[12], which makes it an ideal candidate against cell-to-cell transmission. In a previous study, we identified 10-1074 as one of the most potent bNAb against in vitro cell-associated SHIV$_{162P3}$ infection mediated by semen-infected cells[43]. In addition, 10-1074 was shown to protect macaques from a cell-free viral challenge and to suppress viremia in both NHPs[31,44] and HIV-1 infected patients[39,45,46]. Here, we show that 10-1074 bNAb, formulated for mucosal administration, protected macaques against vaginal cell-associated SHIV infection.

## Results

### In vitro assessment of the cell's infectivity and 10-1074 inhibition of cell-associated transmission

We first established a stock of viable splenic mononuclear cells to be used as challenge inoculum in the efficacy study. Three cynomolgus macaques (CMs; SD01, SD02, and SD03) were intravenously exposed to the R5-tropic virus SHIV$_{162P3}$ and sacrificed between day 10 (*n* = 2) and 13 (*n* = 1) after exposure to collect the spleen (Fig. 1A). The blood viral RNA (vRNA) load at the time of euthanasia was $7.9 \times 10^4$, $4.3 \times 10^6$ and $1.5 \times 10^6$ copies/mL, respectively (Fig. 1B) and was mirrored by the proviral DNA load detected in splenocytes (Table 1). The specific cellular composition of the prepared stock of splenocytes and the gating strategy are reported in Table S1 and Figure S1A and B, respectively. The infectivity of the splenocytes was titrated in vitro using both PBMCs (Fig. 1C) and TZM-bl (Fig. 1D) as target cells and a dose-dependent cell-to-cell transfer of SHIV$_{162P3}$ was observed. In agreement with our previous report[43], 40,000 splenocytes from the three donors were sufficient to allow viral transmission to PBMCs, whereas between 100,000 and 150,000 splenocytes were necessary to infect TZM-bl cells (Fig. 1C and D).

We then assessed the capability of the anti-V3 bNAb 10-1074 to inhibit SHIV$_{162P3}$ transmission by the three stocks of infected splenocytes in a PBMC-based neutralization assay. Coculture experiments were carried out in the presence of decreasing amounts of the Ab until 100% viral infectivity was regained. 10-1074 inhibited SHIV$_{162P3}$ cell-to-cell transmission with a mean IC90 value of $1.52 \pm 0.2$, $1.1 \pm 0.06$ and

$0.32 \pm 0.02 \,\mu g/mL$, when using splenocytes from SD01, SD02, and SD03, respectively. A similar inhibitory effect was observed when splenocytes from two different macaques were combined in a 1:1 ratio (Table 2).

### Stability and pharmacokinetics of the 10-1074 Ab containing gel

Before investigating the pharmacokinetic behavior in the vaginal mucosa of the 10-1074 Ab prepared in Hydroxyethylcellulose (HEC) gel at a concentration of 5 mg/g, we assessed gel stability after 8 weeks of storage at 4 °C and −20 °C. For this purpose, the binding of 10-1074 HEC gel to the HIV-1 YU-2 gp120 wild-type protein (gp120$_{wt}$) or the mutated YU-2 gp120-N332A protein (gp120$_{N332A}$) was assessed in an ELISA assay. Storage conditions did not induce modification in the binding of the 10-1074 HEC gel to the gp120$_{wt}$ protein compared with the monoclonal 10-1074 IgG antibody used as a positive control. As expected, no binding to gp120$_{N332A}$ protein was observed (Figure S2A). In addition, the activity of the 10-1074 HEC gel, when assessed in a TZM-bl-based neutralization assay against three HIV-1 strains known to have a different sensitivity to 10-1074 Ab (i.e. YU-2, TRO11, and PVO.4), was fully preserved (Figure S2B and C). These results fostered us to store the 10-1074 gel at 4 °C during the eight weeks of the challenge.

For the pharmacokinetics study, two Depo-Provera-pretreated female CMs were treated with 2 mL of 10-1074 HEC gel at a concentration of 5 mg/g. Cervicovaginal fluids (CVFs) and blood were collected at different time points (baseline, 1, 2, 4, 6, 24, 48, and 72 h) (Fig. 2A) and Ab concentrations were measured by ELISA against the YU-2 gp120$_{wt}$ protein (Fig. 2B). The amount of active 10-1074 remaining in the CVFs was calculated as a function of time. CVF collected between 1 h and 6 h after treatment contained mean Ab concentrations above 4000 $\mu g/mL$ (i.e. > 10.000-fold the IC50 against cell-free HIV-1$_{YU2}$ and cell-associated SHIV$_{162P3}$), dropping to less than 600 $\mu g/mL$ after 24 h in both animals (Fig. 2B). Levels of 10-1074 concentrations in serum at the same time points remained below detection limits (40 ng/mL). Finally, we evaluated CVFs for their inhibitory potency against cell-free SHIV$_{162P3}$ in a TZM-bl-based neutralization assay (Fig. 2C), showing a time-dependent loss of inhibitory activity which become particularly evident for CVFs collected between 24 and 72 h compared to earlier time points. IC50 values, i.e. CVF dilutions at which TZM-bl infection was reduced by 50%, are shown in Table S2.

### 10-1074 HEC gel protects from infection after cell-associated vaginal SHIV$_{162P3}$ exposure

For the vaginal challenge study, Depo-Provera-pretreated female CMs were subjected to weekly vaginal applications of 2 mL HEC gel containing no Ab (placebo gel, *n* = 6) or 5 mg/g of the 10-1074 bNAb (10-1074-gel, *n* = 6). We ensured complete absorption of the HEC gel into the vaginal vault, and its successful utilization in previous protection studies by our group[26,47–49] indicated no safety concerns. One hour later animals were exposed intravaginally to $10^7$ SHIV$_{162P3}$.infected splenocytes. The experimental design of the study is shown in Fig. 3. A mix of splenocytes from two donors (1:1 ratio) was used at each challenge. The challenge schedule and amount of inoculated SIV DNA per challenge are described in Table 3.

All except one (CM01) of the six animals receiving the placebo gel displayed high viral titers after two to seven challenges, whereas five out of six animals treated with the 10-1074 gel were fully protected after eight challenges (Fig. 4A and Table S3). A Kaplan-Meier analysis showed that 10-1074 treatment resulted in a significant difference when compared to the control group (80% of protection, *p* = 0.0227, Fig. 4B). In the Ab-treated group, the five protected animals exhibited no detectable viremia throughout the whole study. Macaque CM07, which was infected after five challenges, displayed a lower viremia relative to infected control animals when considering both the vRNA peak (56,336 *vs* a mean of $1.8 \times 10^6$ copies/mL, respectively) and the area under the curve (AUC) of viremia between weeks 1–15 (316,848 *vs*

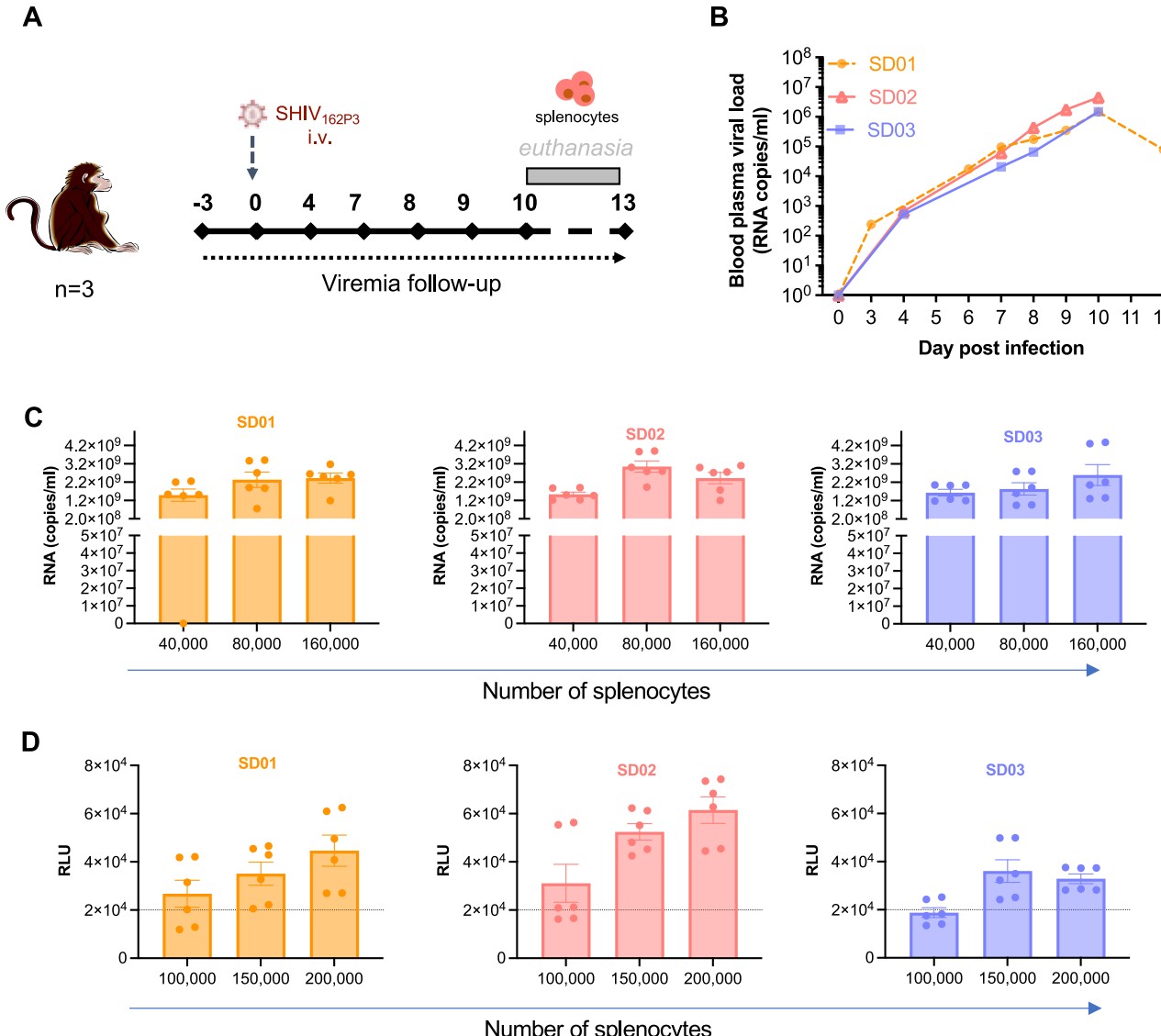

**Fig. 1 | Establishment of SHIV$_{162P3}$-infected splenocyte stocks and in vitro assessment of cell-mediated infection. A** Study design to produce a stock of SHIV$_{162P3}$ infected splenocytes following intravenous (i.v.) virus challenge. **B** Follow-up of blood vRNA load in three macaques intravenously infected with SHIV$_{162P3}$ up to the day of euthanasia. SD indicates splenocyte donor. **C** Titration of splenocytes (40,000–160,000 cells) in coculture with 120,000 PBMCs. Viral production was detected by RT-qPCR 7 days postcoculture. **D** Titration of splenocytes (100,000–200,000 cells) in coculture with $10^4$ TZM-bl cells. Infection was recorded as Relative Light Unit (RLU) at 48 h post co-culture after subtraction of the RLU of the control wells (TZM-bl only). Each bar represents the mean and standard error of the mean (SEM) of two independent experiments performed in triplicate. Each color represents a different animal. Source data are provided as a Source Data file.

a mean of 16 ×$10^6$ copies/mL, respectively) (Table 4). In this animal, the viral load become undetectable at week 10 and thereafter remained below the lower limit of quantification (LOQ, 111 copies/mL) till the end of the study. However, a similar AUC was observed in the CM02 control animal (Table 4).

As expected with SHIV$_{162P3}$[26,49], a proportion of the infected animals eventually controlled viremia and displayed fluctuating viral kinetics. A transient lymphopenia was observed in all animals at the time of first detectable vRNA, which was particularly strong in CM04 (Table S4). Protection in treated animals could not be attributed to specific major histocompatibility complex (MHC) protective alleles nor to differences in age or weight at the beginning of the study (Table S5).

In addition, we confirmed that none of the protected animals had a detectable SIV Gag DNA in lymph nodes collected at week 15 after the first challenge (Fig. 4C) and in PBMCs through the study follow-up (Fig. 4D). The higher amount of integrated SIV DNA in both lymph nodes and PBMCs was detected in macaque CM04 which also displayed the highest peak viremia in the blood (Fig. 4C and D).

Quantification of viral reservoirs seeding in tissues at necropsy, when the blood vRNA was below the LOQ in all except two animals (CM04 and CM05), demonstrated virus persistence in inguinal lymph nodes, and/or iliac lymph nodes, and spleen from CM02, CM04, CM05, and CM06 control animals as well as in inguinal lymph nodes from the

### Table 1 | SIV Gag DNA content of SHIV$_{162P3}$-infected spleen cells

| Splenocyte donor | SIV DNA (copies/$10^6$ cells) |
| --- | --- |
| SD01 | 1.15 ×$10^4$ |
| SD02 | 4.02 ×$10^4$ |
| SD03 | 8.3 ×$10^4$ |

Ab-treated CM07 macaque (Table 5). Conversely, the virus was below the lower limit of detection (LOD = 46 copies/10⁶ cells) in the female reproductive tract (cervix, uterus, vagina, and fallopian tubes) from both treated and control animals.

Finally, to demonstrate that infection occurred through SHIV associated with inoculated cells and not through free viral particles contaminating the inoculum, we collected the supernatant of the final wash of splenocytes before each of the eight inoculations of cells to macaques. The amount of viral RNA in those supernatants was below the lower LOD, and they did not transmit the infection to the highly susceptible TZM-bl cells in vitro (Figure S3). To further corroborate the inhibitory effect of the 10-1074-containing gel on cell-associated virus as opposed to cell-free virus produced by infected cells, we conducted in vitro culturing of 10⁶ SHIV$_{162P3}$-infected splenocytes for a period of up to 24 h. The culture was conducted both in the absence and presence of CVFs to simulate the vaginal environment. At multiple time points (1, 2, 6, and 24 h) during the culturing process, the amount of viral RNA detected in the supernatant remained below the lower LOD (Table S6).

### Detection of adaptive immune responses in SHIV-infected animals

To assess the development of adaptive immunity in treated and control animals, we measured Env-binding IgG titers in serum during follow-up and T-cell responses to Env and Gag at the time of animal euthanasia. The five protected animals treated with the 10-1074 HEC gel, as well as the CM01 control animal, remained seronegative throughout the duration of the study, whereas all animals with detectable viremia developed anti-Env specific antibodies between 4 and 8 weeks after the first challenge (Fig. 5A). T-cell responses were weak but detectable only in viremic animals, including the five treated with the placebo gel and the 10-1074 gel-treated animal CM07 (Fig. 5B

**Table 2 | Inhibitory concentration (IC) 50, 75 and 90 of 10-1074 in a PBMC-based neutralization assay using the different stocks of splenocytes as donor cells**

| Splenocyte donor | 10-1074 bNAb (µg/mL)ᵃ | | |
|---|---|---|---|
| | IC50 | IC75 | IC90 |
| SD01 | 1.07 ± 0.2 | 1.33 ± 0.1 | 1.52 ± 0.2 |
| SD02 | 0.2 ± 0.01 | 0.59 ± 0.05 | 1.1 ± 0.06 |
| SD03 | 0.18 ± 0.04 | 0.26 ± 0.12 | 0.32 ± 0.2 |
| SD03 + SD02 | 0.42 ± 0.03 | 0.54 ± 0.01 | 1.03 ± 0.13 |
| SD03 + SD01 | <0.062 | 0.3 ± 0.11 | 0.43 ± 0.05 |

ᵃMean ± standard deviation from three independent experiments.

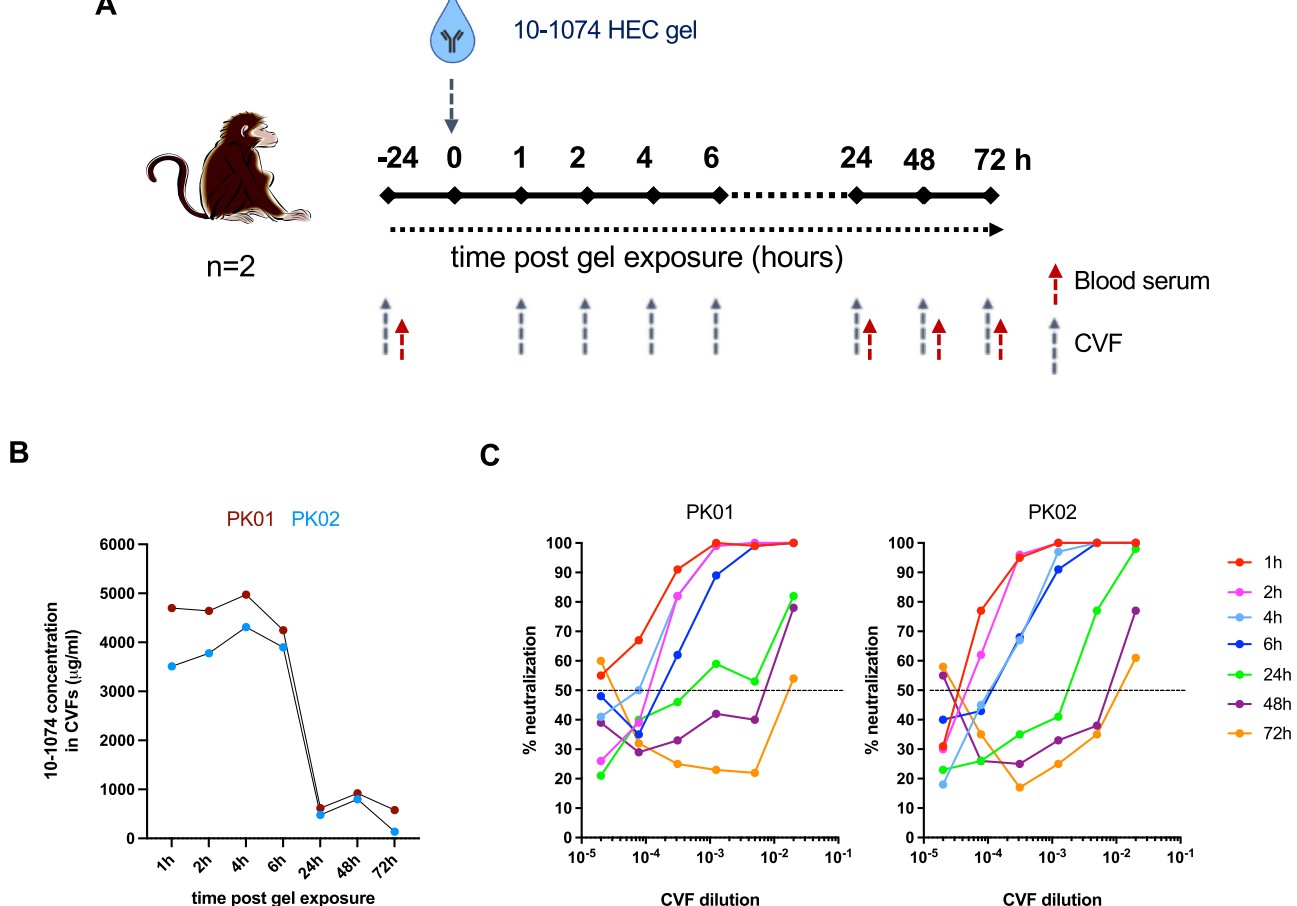

**Fig. 2 | Pharmacodynamics of 10-1074 HEC gel treatment in cervicovaginal fluids. A** Study design to determine the pharmacodynamics following the application of 2 mL of the 10-1074 Ab formulated in a 1.6% hydroxyethylcellulose (HEC) gel at 5 mg/g inside the vaginal vault of two macaques. Cervicovaginal fluids (CVFs) and blood serum were collected before and after gel inoculation at the time points indicated in the figure. **B** 10-1074 Ab concentration was detected in CVFs collected at several time points after 10-1074 gel application in two macaques, #PK01 (red curve) and #PK02 (blue curve). **C** Neutralization of SHIV$_{162P3}$ by CVFs collected at different time points following vaginal application of the 10-1074 gel was assessed in a TZM-bl assay system. Mean values from a representative experiment performed in triplicate are shown. The IC50, i.e., the CVF dilution able to decrease by 50% the infection of the cells, was defined (dotted line). Source data are provided as a Source Data file.

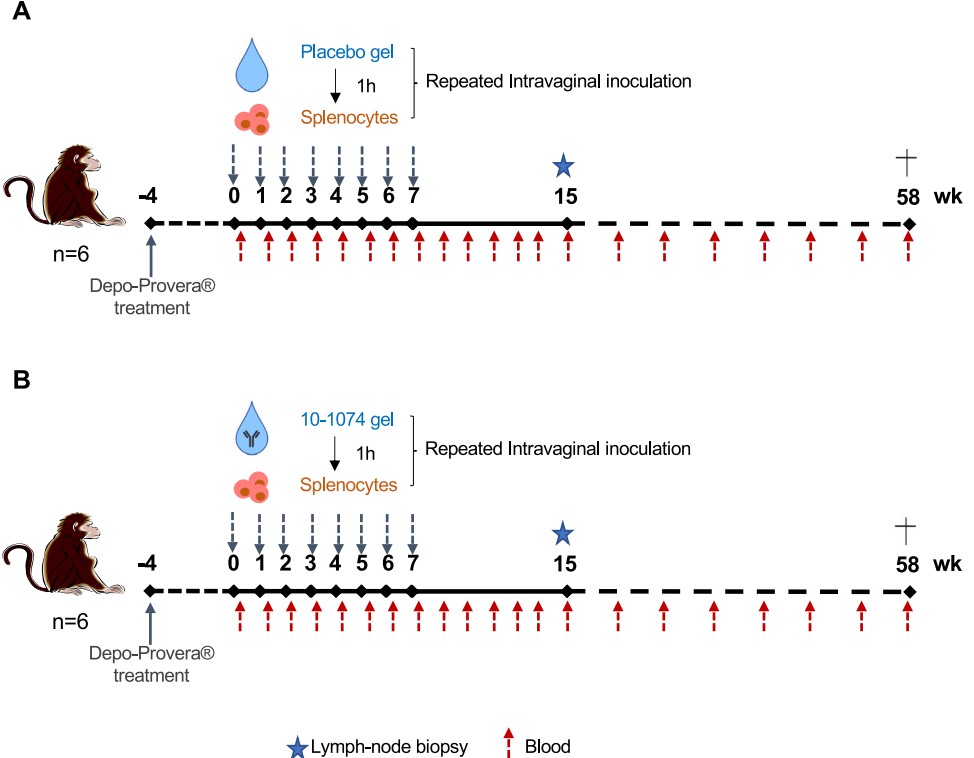

**Fig. 3 | Experimental design of the protection study.** Twelve female cynomolgus macaques were treated with Depo-Provera. Four weeks (wk) later, 2 mL of either (**A**) a placebo HEC gel (no Ab, *n* = 6, control group) or (**B**) 10-1074 Ab (*n* = 6, treated group) formulated in HEC gel at the concentration of 5 mg/g (10 mg of Ab/treatment/animal) were applied in the vaginal vault. One hour later, all animals were exposed to $10^7$ SHIV$_{162P3}$-infected splenocytes. The treatment was repeated weekly for up to 8 challenges or until animals were confirmed viremic on two consecutive weeks. Blood was collected at baseline (before the first challenge), weekly up to week 12 after the first challenge, then at weeks 15, 19, 23, 31, 39, 43, 52, and 58. 15 weeks after the first challenge, inguinal lymph node biopsies were collected. The dagger indicates the time of the necropsy.

and C). These data suggest that SHIV$_{162P3}$-cell-associated infection fosters B and T cell responses in this model.

## Discussion

Passive immunization using recombinant monoclonal antibodies capable of neutralizing diverse HIV-1 isolates is a valuable approach for achieving long-acting pre-exposure prophylaxis. Because of their favorable safety and pharmacokinetic profiles, bNAbs may offer advantages over traditional antiretroviral therapy for both the prevention and treatment of HIV-1 infection and considerable efforts are ongoing to induce bNAb through vaccination. SHIV infection of macaques allows for the preclinical assessment of the efficacy, safety, and immunological impact of bNAbs in a setting that closely mimics HIV infection in humans. The development of relevant animal models evaluating the prevention of infection with both cell-free and cell-associated viruses is paramount for designing effective immune-based prophylactics.

Here, using a modification of our previously published model of in vivo cell-associated SIV transmission in NHPs[5], we show that mucosal administration of the bNAb 10-1074 conferred significant protection against repeated vaginal exposure to SHIV$_{162P3}$. The data presented here are consistent with those of previous studies demonstrating the capacity of 10-1074 to protect Rhesus macaques from cell-free SHIV$_{AD8EO}$ rectal challenge in vivo[23] and, when used in combination with 3BNC117 and PGT121, to suppress viremia to an undetectable level in SHIV$_{162P3}$ and SHIV$_{AD8}$ chronically infected animals[44,50]. Furthermore, our observation that 10-1074 provided robust protection in vivo against repeated cell-associated SHIV$_{162P3}$ challenge is consistent with our previous report showing a strong efficacy of the 10-1074 Ab in inhibiting ex vivo cell-to-cell transmission mediated by SHIV$_{162P3}$-infected semen cells[43].

In this study, spleen cells collected at the peak of plasma viremia from SHIV$_{162P3}$-infected macaques were used as a surrogate of semen cells. This experimental model was demonstrated to recapitulate infection mediated by seminal leukocytes because of the similar cellular composition and activation level by monocytes/macrophages, CD4 +, and CD8 + T cells observed in the spleen and semen cells[43]. Additionally, our cell-associated intravaginal challenge model mimics HIV exposure in women, and our results demonstrating a significant reduction in vaginal SHIV infection risk among macaques could inform dose selection for the development of bNAbs as topical pre-exposure prophylaxis candidates for use in clinical settings. The success of such strategies that aim to prevent HIV-1 transmission via sexual exposure may rely on the quantity and functionality of bNAbs present in the secretions and tissues of the genital tract[51,52], as well as on the presence of infected cells capable to interact with the epithelial layer and rapidly infiltrate the recipient's mucosa[4,5].

Previous attempts to block cell-associated SHIV transmission by PGT121 bNAb following intravenous challenge of infected cells were

**Table 3 | Splenocytes preparation for each challenge and amount of inoculated SIV Gag DNA copy number per challenge**

| Challenge | Splenocyte donor | Amount of SIV DNA/donor (copies/5×10⁶ cells/donor) | Total amount of SIV DNA (copies/10⁷ cells/challenge) |
|---|---|---|---|
| 1st - 3rd- 5th - 7th | SD03 + SD01 | 415,000 + 57,500 | 472,500 |
| 2nd - 4th - 6th- 8th | SD03 + SD02 | 415,000 + 210,000 | 625,000 |

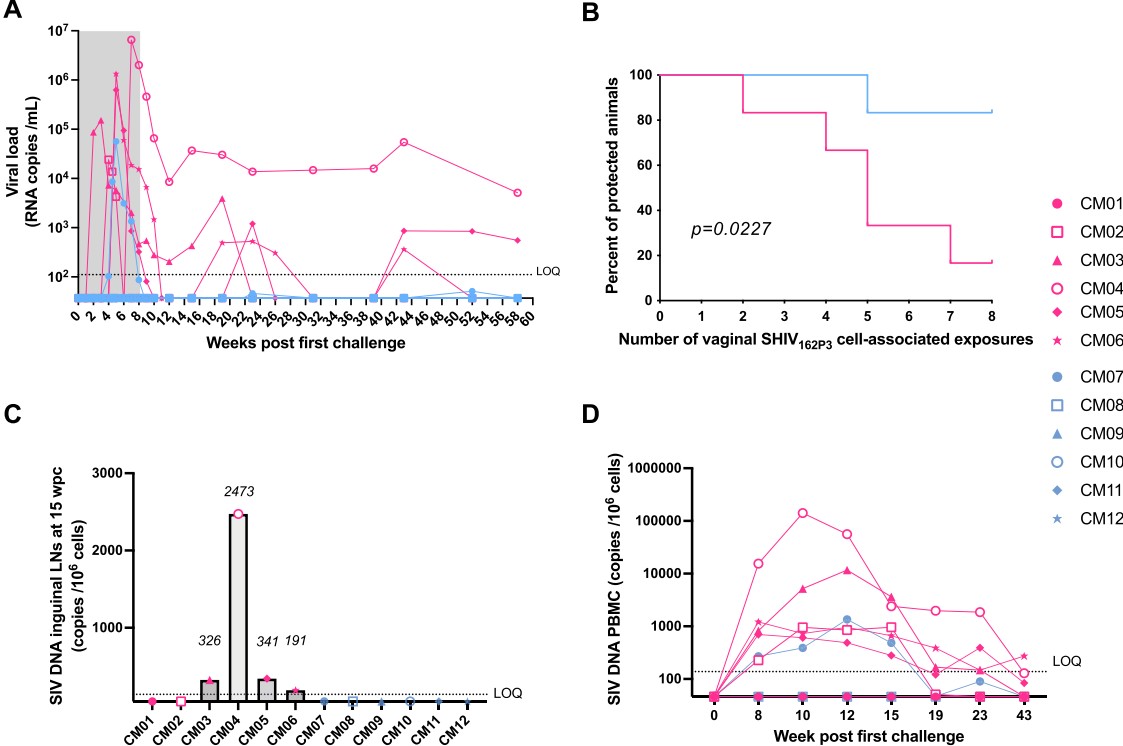

**Fig. 4 | Protection by a 10-1074 HEC gel during repeated vaginal SHIV$_{162P3}$ cell-associated challenges. A** Follow-up of plasma vRNA load from macaques treated with the placebo gel ($n = 6$, control group, pink symbols) or with the 10-1074 gel ($n = 6$, treated group, blue symbols) and intravaginally exposed to $10^7$ SHIV$_{162P3}$-infected splenocytes. The gray area indicates the eight weeks of challenges. The dotted line indicates the limit of quantification (LOQ = 111 copies/mL) for the viral load assay performed. Group colors are consistent throughout the manuscript. **B** The number of animals remaining uninfected is plotted against the number of

SHIV$_{162P3}$ cell-associated viral challenges. The Kaplan-Meier survival curves are significantly different in the placebo gel control group (pink curve) compared to the 10-1074 gel-treated group (blue curve) ($p = 0.0227$; Log-rank (Mantel-Cox) test). **C** SIV Gag DNA copy number detected in inguinal lymph nodes from the twelve macaques is reported for animals with detectable viremia. **D** SIV Gag DNA copies number detected in PBMC collected during the first 43 weeks of the study. The dotted line indicates the assay's quantification limit (LOQ = 138 copies/$10^6$ cells). CM = cynomolgus macaque. Source data are provided as a Source Data file.

described by Parsons et al.[40,41]. Partial protection was observed in animals (i.e., in 3 of 6) infused intravenously with 1 mg/kg of PGT121 and challenged 1 h later with $2.45 \times 10^7$ SHIV$_{162P3}$-infected splenocytes, whereas complete protection was observed using a 5-fold-reduced challenge dose of $5 \times 10^6$ splenocytes. In our study targeting the site of virus exposure on mucosal sites, vaginal application of an HEC gel containing 5 mg/g (i.e. 5 mg/ml) of 10-1074 antibody protected 5 of 6 animals against vaginal challenge with $1 \times 10^7$ SHIV$_{162P3}$-infected

**Table 4 | Blood viral load detected in the placebo and in the 10-1074 Ab treated group**

| Animal ID | Group | peak plasma viral load (copies/mL) | AUC week 0–15 (copies/mL) |
|---|---|---|---|
| **CM01** | **Placebo gel** | <LOD | <LOD |
| **CM02** | **Placebo gel** | 24,099 | 212,102 |
| **CM03** | **Placebo gel** | 151,883 | 1,822,455 |
| **CM04** | **Placebo gel** | 6,562,634 | 64,445,176 |
| **CM05** | **Placebo gel** | 629,939 | 5,077,402 |
| **CM06** | **Placebo gel** | 1,327,808 | 10,007,285 |
| **CM07** | **10- 1074 gel** | 56,336 | 316,848 |
| **CM08** | **10- 1074 gel** | <LOD | <LOD |
| **CM09** | **10- 1074 gel** | <LOD | <LOD |
| **CM10** | **10- 1074 gel** | <LOD | <LOD |
| **CM11** | **10- 1074 gel** | <LOD | <LOD |
| **CM12** | **10- 1074 gel** | <LOD | <LOD |

*AUC* area under the curve.
*LOD* lower limit of detection, 37 copies/mL.

splenocytes. The pharmacokinetic study revealed a mean Ab concentration in CVFs of 4 mg/ml up to 6 hours after gel application, which is equivalent to >10,000-fold the in vitro IC50 against cell-associated SHIV$_{162P3}$. These results offer valuable insights into the Ab concentration required for potential human trials to achieve optimal protection, subject to appropriate scaling methods. Subsequently, between 24 to 72 h, the concentration declined by 6-fold but remained detectable in CVFs. These results align with previous studies involving vaginally applied gel-formulated molecules such as bNAbs, CD4 mimetic, and recombinant chemokine analogs[26,47,49,53], which also exhibited sustained activity in the vaginal environment for a few hours. Together those data indicate that the administration route, the site of exposure to the virus, and the dose should be considered when evaluating prophylactic strategies against cell-associated HIV-1 transmission. To achieve full protection in humans, it may be necessary to attain a specific combination of systemic and mucosal concentrations of bNAbs, which may differ depending on whether the dosing is oral or topical. The utilization of preclinical models, such as the one illustrated in this study, may provide valuable insight in this regard.

In this study, the infection observed in one 10-1074 treated macaque (CM07) following the fifth challenge, was cause for concern. It is plausible to assume that a lower mucosal Ab concentration was occasionally achieved in this animal however, we could not ascertain this hypothesis because cervicovaginal fluids or vaginal biopsies during the challenge phase were not available. Modest decrease in peak viral load and rapid control of viremia, along with a delayed seroconversion, was observed in this animal, although a similar viral load kinetics was also observed in one control animal. Higher Ab concentration may be required to ensure full protection. Also, combining

**Table 5 | SIV Gag DNA content in organs collected at the end of the study**

| Animal ID | Group | Inguinal LN (left) | Inguinal LN (right) | Iliac LN (left) | Iliac LN (right) | Spleen |
|---|---|---|---|---|---|---|
| SIV DNA copies/10^6 cells | | | | | | |
| CM01 | Placebo gel | <LOD | <LOD | <LOD | <LOD | <LOD |
| CM02 | Placebo gel | 60 | <LOD | <LOD | <LOD | <LOD |
| CM03 | Placebo gel | n.a. | <LOD | n.a. | <LOD | <LOD |
| CM04 | Placebo gel | 1428 | 1966 | 1138 | 2429 | 350 |
| CM05 | Placebo gel | 102 | 99 | 98 | 106 | 241 |
| CM06 | Placebo gel | 211 | 117 | 104 | 92 | 324 |
| CM07 | 10- 1074 gel | <LOD | 1332 | <LOD | <LOD | <LOD |
| CM08 | 10- 1074 gel | <LOD | <LOD | <LOD | <LOD | <LOD |
| CM09 | 10- 1074 gel | <LOD | <LOD | <LOD | <LOD | <LOD |
| CM10 | 10- 1074 gel | <LOD | <LOD | n.a. | <LOD | <LOD |
| CM11 | 10- 1074 gel | <LOD | <LOD | <LOD | n.a. | <LOD |
| CM12 | 10- 1074 gel | <LOD | <LOD | <LOD | <LOD | <LOD |

*LOD* lower limit of detection, 46 copies/10^6 cells.
n.a. = not available.

bNAbs is likely to enhance the observed protective effects, as their enhanced potency and broader efficacy may enable dose-sparing effects whilst protecting against the possible emergence of resistant variants.

Conversely, one of the control macaques (CM01) was completely protected from cell-associated SHIV transmission after eight vaginal exposures. In our previous study, two macaques exposed to $10^7$ splenocytes carrying $4 \times 10^5$ DNA copies/mL become infected after the first and the sixth challenge - albeit in the absence of a placebo gel[43]. Here, using approximately 10 times fewer infected cells, 5 out of 6 macaques become infected. We acknowledge that the dose of cell-associated SHIV infecting 50% of the animals (AID50) was not estimated in this study and that in vivo intravaginal viral titrations would inform future challenge studies using infected cells. In previous studies of intravenous cell-associated SHIV$_{162P3}$ transmission[40,41], 100% of macaques become infected after a single exposure to amounts of infected splenocytes similar to the one used here, suggesting that mucosal transmission by the cell-associated virus might be more difficult to attain compared to intravenous exposure. This hypothesis is sustained by epidemiological data estimating that the likelihood of acquiring HIV from an infected source per act is approximately 8 times higher with needle-sharing/injection drug use (63/10,000 exposures) than with receptive vaginal intercourse (8/10,000 exposures)[54]. Finally, we cannot exclude the possibility of intrinsic resistance of CM01 to SHIV$_{162P3}$ infection, although it was not a carrier of protective MHC alleles.

The repeated low-dose cell-associated viral challenge led to productive infection in 5 out of 6 animals within the control group. Among these, four animals demonstrated eventual viremia control, with only two macaques showing detectable viral load at 1 year post-infection. The noteworthy high rate of virologic control observed can be attributed to the utilization of a low infection dose. Although direct comparisons are limited due to the unique nature of our study, being the first to assess cell-associated SHIV transmission, these findings are consistent with previous reports from both our group and others, consistently demonstrating a high frequency of viremia control in cynomolgus macaques following cell-free SHIV$_{162P3}$ infection[26,49,55,56].

The AMP study of VRC01's ability to prevent HIV-1 infection indicated that in vivo it may be necessary to achieve higher concentrations of bNAbs than what is indicated by in vitro sensitivity studies[57]. It is plausible to assume that the inability of in vitro studies to fully predict efficacy in humans might be due, at least in part, to the overlooked cell-associated transmission, which is not systematically assessed in vitro and in preclinical models. It has indeed been observed that the effectiveness of antibodies is typically diminished during HIV-1 transmission via cell-to-cell contact, as opposed to infection through cell-free virus[8,10,12–14,43]. This phenomenon may be ascribed to differences in the envelope protein's conformation or accessibility on the surfaces of virions and cells. A major obstacle to bNAb immunotherapy is the ability of the cell-associated virus to evade the Ab neutralization[1]. In addition, non-neutralizing Fc-mediated antibody functions are important for bNAb-conferred protection from the cell-free virus in non-human primates[26,58], although they were described as partially redundant in a cell-associated transmission model[41]. Further studies might help dissect how Fc-dependent effector functions can be optimized to improve the protective efficacy of bnAbs against cell-associated HIV-1. Studies results might be depending on the antibody used and the differential abilities of HIV-1 strains to escape bNAbs. Understanding the mechanism of action and the interaction of infected cells with various bNAbs is crucial, especially since several of these antibodies, including 10-1074, have undergone clinical evaluation in recent years. Furthermore, we recognize the significance of evaluating microbicide efficacy against cell-associated transmission occurring across the rectal mucosa in both preclinical and clinical investigations.

Our study provides strong evidence for the effectiveness of 10-1074 in controlling cell-associated HIV-1 transmission in non-human primates and supports the ongoing development and evaluation of bNAbs as a treatment option for individuals living with HIV. Additionally, our findings underscore the importance of assessing the efficacy of passive immunization approaches against cell-associated virus, which is a critical consideration in the development of an HIV-1 preventive vaccine.

## Methods
### Experimental design
Cynomolgus macaques (CMs, *Macaca fascicularis*) originating from Mauritian AAALAC-certified breeding centers were housed in the IDMIT facilities (CEA, Fontenay-aux-Roses) under BSL-3 containment (facility authorization #D92-032-02, Préfecture des Hauts de Seine, France) and in compliance with the European Directive 2010/63/EU, French regulations, and the Standards for Human Care and Use of Laboratory Animals. A total of 17 CMs were included in this study. Animal characteristics are described in Table S5. Three males were involved in splenocyte production, two females in the longitudinal PK study, and twelve females in the efficacy study (six in the placebo gel control group, and six in the 10-1074 HEC gel treated group). All animals tested negative for SIV, STLV (simian T-lymphotropic virus), herpes B virus, filovirus, SRV-1, SRV-2, and measles before enrolling in the study. MHC haplotypes were determined by microsatellite analyses[59]. The group size for virus challenge experiments was determined based on previously published results and statistical analysis[5,26]. Experimental procedures were performed under anesthesia using ketamine (5 mg/kg) and medetomidine (0.042 mg/kg).

### Cells, reagents, and viruses
TZM-bl cells were obtained from the NIH AIDS Research and Reference Reagent Program (NIH ARRRP, catalog number ARP-8129) and cultivated in DMEM containing 10% heat-inactivated FCS and antibiotics. Human PBMCs from healthy donors were prepared from buffy coat and then cultivated in RPMI containing 10% FCS, 1% PSN, 5 μg/ mL PHA (Sigma-Aldrich), and 5 units/mL IL-2 (Roche) for three to four days before use. Buffy coat from healthy human donors was purchased from the Établissement Français du sang (EFS). PBMCs were isolated and collected using Lympholyte cell-separation media (Cedarlane) by

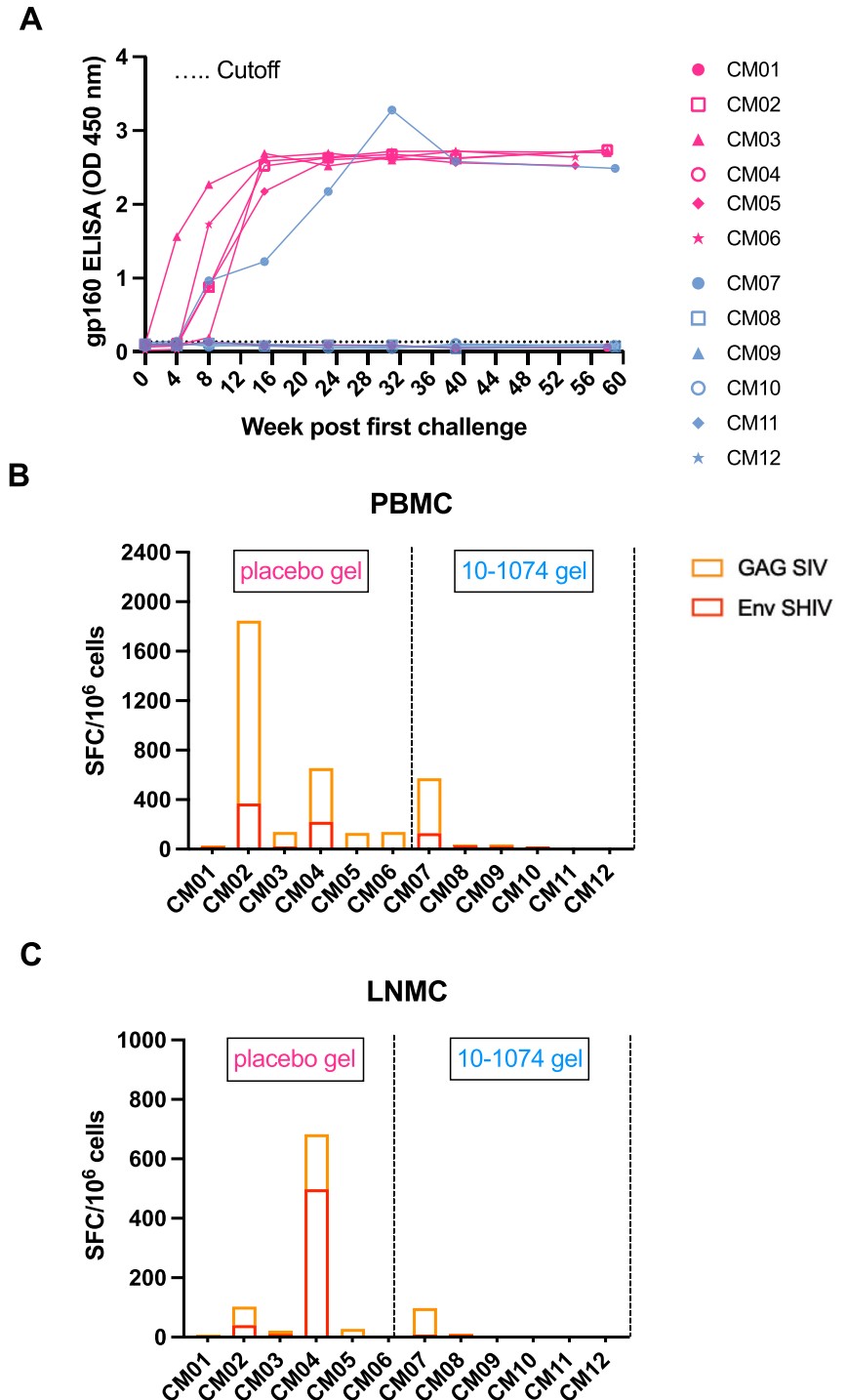

**Fig. 5 | 10-1074 Ab treatment does not modulate adaptive immunity.**
**A** Seroconversion of macaques challenged with cell-associated SHIV$_{162P3}$. An enzyme-linked immunosorbent assay (ELISA) was used to screen serum samples from the placebo gel group (pink curves) and 10-1074 gel group (blue curves) and challenged with cell-associated SHIV$_{162P3}$ for the presence of antibodies against the gp160 envelope protein. The dotted line represents the cutoff optical density (OD) for the ELISA, which was defined as three times the OD of gp160-coated wells with a SHIV$_{162P3}$-uninfected plasma sample. **B, C** T-cell responses to SIVmac241 Gag antigen and SHIV$_{162P3}$ Env peptide pool were measured by IFNγ ELISPOT. T-cell responses were quantified in PBMC (**B**) and lymph-nodes mononuclear cells (LMNCs) (**C**) at the time of death. Concanavalin A was used as positive stimulation control. Group colors are consistent throughout the manuscript. CM = cynomolgus macaque. Source data are provided as a Source Data file.

centrifugation, aliquoted, and frozen at -80 °C for cell-to-cell transmission and neutralization assays.

YU-2 gp120 proteins (gp120$_{wt}$ and gp120$_{N332A}$) were produced by transient transfection of Freestyle™ 293-F cells (Thermo Fisher Scientific) using polyethyleneimine (PEI)-precipitation method, purified by

chromatography using the Ni Sepharose® Excel Resin (GE Healthcare) according to manufacturer's instructions.

Recombinant IgG antibodies, i.e. human HIV-1 anti-V3-glycan 10-1074 IgG[60] and non-HIV-1 antibody mGO53 IgG[61] used as a negative control, were produced by cotransfection of Freestyle 293-F cells

(Thermo Fisher Scientific) using PEI precipitation method as previously described[62,63] and purified by affinity chromatography using protein G Sepharose 4 fast flow beads (GE Healthcare).

SHIV$_{162P3}$ virus used for in vivo infection and efficacy studies was obtained from the NIH ARRRP. The infectious molecular clone (IMC) of the SHIV$_{162P3}$ virus used for in vitro studies of Ab inhibitory activity was produced by transfection of HEK-293T cells with 12 μg of the pNL-LucR.T2A-SHIV-SF162P3.5.ecto plasmid (kindly provided by Dr. C. Ochsenbauer, University of Alabama at Birmingham, USA), using Fugene 6 transfection reagent (Roche, Indianapolis, IN, USA). HIV-1 YU-2, PVO.4, and TRO11 pseudoviruses were produced by co-transfecting HEK-293T cells with 10 μg of pYU-2 or pPVO.4 or pTRO11 env expression vector (from the NIH AIDS reagent program) and 20 μg of backbone plasmid (pSG3 delta env vector)[64]. Pseudovirus-containing culture supernatants were harvested 2 days after transfection, filtered (0.45 μm), and stored at −80 °C in 1-ml aliquots.

### Splenocyte stocks preparation and phenotypic characterization

Three male CMs (SD01, SD02, and SD03) were infected by intravenous inoculation with a single dose of 5×10$^2$ 50% tissue infectious doses (TCID50) of SHIV$_{162P3}$. The animals were euthanized at the peak of viremia (between days 10 and 13 post-infection) and spleen cells were mechanically dissociated and isolated by density-gradient separation with Lympholyte cell separation media (Cedarlane). Aliquots of cells were frozen in 10% fecal calf serum in dimethyl sulfoxide (DMSO).

Staining was performed on thawed splenocytes. Amine-reactive blue dye (Live/dead Fixable, Life Technologies) was used to assess cell viability and exclude dead cells from the analysis. Cells were stained with monoclonal antibodies by incubation for 30 min at 4 °C, washed in PBS/10% FCS and fixed in commercial fixation solution (CellFIX, BD Biosciences).

Two-different antibody panels were used: panel 1, targeting lymphocyte populations and T cell activation markers, and panel 2, targeting monocytes, dendritic cells, and neutrophils. A list of antibodies used for staining the splenocytes is shown in Table S7. Data were acquired on an LSRFortessa (BD Biosciences) using Diva software (BD Biosciences) and analyzed with Flowjo v10 (BD Biosciences).

### Viral DNA extraction from splenocytes, quantification, and normalization

DNA was extracted from frozen cells using the NucleoSpin Tissue XS kit (Macherey-Nagel), according to the manufacturer's instructions. SIV DNA was quantified in duplicate by real-time PCR Taqman assay using the Platinum® Quantitative PCR SuperMix-UDG kit (Thermo Fisher Scientific) with SIV-gag primers and probe, as described elsewhere[65]. The lower limit of detection (LOD) was 46 copies per 10$^6$ cells. The reaction, data acquisition, and analysis were performed with the CFX96 Touch Real-time PCR Detection System (Bio-Rad).

### Assessment of cell-to-cell transmission in the TZM-bl assay

Cell-to-cell transmission was assessed by co-culturing each stock of splenocytes (100,000–200,000 cells) with 10$^4$ TZM-bl, as previously described in[43]. The co-culture was performed in 96 flat-well plates in DMEM medium containing 10% heat-inactivated FCS and antibiotics for 48 h at 37 °C in the absence of DEAE dextran. The infection was estimated by lysing the cells and measuring the production of the luciferase reporter gene in TZM-bl upon the addition of firefly luciferase substrate (Promega, Madison Wisconsin, USA).

### Assessment of cell-to-cell transmission and neutralization in the PBMC assay

Each stock of splenocytes (input 40,000 – 160,000 cells) was titrated to define the number of cells allowing efficient cell-to-cell transmission. 10-1074 IgG bNAb was used at a starting concentration of 5 μg/mL and four three-fold dilutions were tested. Cells were incubated for 1 h with the Ab before co-culture with 120,000 PHA-IL-2-activated PBMCs from two donors.

The co-culture in 96 round-bottom well plates was performed in RPMI medium containing 10% heat-inactivated FCS, antibiotics, and 20 units/mL IL-2 at 37 °C. After seven days of culture, the supernatant was collected to detect viral replication by RT-qPCR (Superscript III platinum one-step qPCR system), according to the manufacturer's recommendations. Neutralization titers were defined as previously described in[43]. The IC50, IC75, and IC90 were defined as the concentration of Abs able to decrease the percentage of infected cells by 50%, 75%, and 90%, respectively, and were calculated using a linear interpolation method[66].

### 10-1074 Hydroxyethylcellulose (HEC) gel preparation and stability studies

A citrate buffer solution (5 mM, pH 5.8) was prepared by dissolving citric acid monohydrate (231 mg) and trisodium citrate dihydrate (1116 mg) in about 900 mL of MilliQ® water. The volume of the solution was then brought up to 1 L with the addition of MilliQ® water and mixed with a magnetic stirrer. The solution was filtrated through a 0.22 μm filter.

The placebo gel was prepared by successively adding gelling polymer HEC (216 mg, final concentration 1.5 wt%) and citrate buffer solution (10 mL). Then, sorbic acid (12 mg, final concentration 0.1 wt%) and glycerol (300 mg, final concentration 2.5 wt%) were added. The volume of the vial was then brought up to 12 mL with the addition of citrate buffer solution. The final formulation was mixed with a magnetic stirrer for at least 4 hours until the complete dissolution of HEC.

The 10-1074 gel was prepared by adding to a vial the antibody (750 μL of Ab at a concentration of 20 mg/mL). Then, the weight was brought up to 3 g by adding HEC. The final concentration of the Ab in the gel was 5 mg/g.

Both the placebo and the 10-1074 HEC gels were stored at 4 °C during the 8-week challenge period and their stability was assessed by ELISA. Briefly, high-binding 96-well ELISA plates (Costar, Corning) were coated overnight with HIV-1 YU-2$_{wt}$ and YU-2$_{N322A}$ Env proteins (125 ng/well in PBS). After washing with 0.05% Tween 20-PBS (PBST), plates were blocked for 2 h with 2% bovine serum albumin and 1 mM EDTA-PBST (blocking solution), washed, and incubated with serially diluted gels and 10-1074 recombinant monoclonal IgG antibody (as control) in PBS. After washing, plates were revealed by the addition of goat HRP-conjugated anti-human IgG (Immunology Jackson ImmunoResearch) and HRP chromogenic substrate (ABTS solution; Euromedex). Experiments were performed using a HydroSpeed microplate washer and Sunrise microplate absorbance reader (Tecan Männedorf), with optical density measurements made at 405 nm (OD405nm).

The neutralization activity of the 10-1074 HEC gel was measured using TZM-bl reporter cells in duplicate as previously described in[67]. IC50 values were calculated using Prism software by fitting duplicate values using the five-parameter sigmoidal dose–response model.

### Pharmacokinetics studies in NHPs

A PK study of the 10-1074 Ab-containing gel was conducted in two naïve female CMs (PK01 and PK02) pre-treated intramuscularly with 30 mg medroxyprogesterone acetate (Depo-Provera, Pfizer) 30 days before the gel application. Two mL of HEC gels containing 10-1074 Ab at 5 mg/g were applied into the vaginal vault with a French catheter connected to a ready-to-use syringe. Before procedures, the animals were anesthetized with a 2.5 mg/kg intra-muscular injection of Zoletil®100 (Virbac, Carros, France). Animals were monitored for product leakage. Blood was collected before and at 24, 48, and 72 h after gel application. CVFs were collected before and at 1, 2, 4, 6, 24, 48, and 72 h after gel application, by placing pre-weighted Weck-Cel sponges

(Beaver Visitec International) into the vaginal vault. Upon removal, sponges were reweighed to calculate the collected vaginal fluid weight, and sponge contents were eluted in 600 µl of PBS containing 0.25 M NaCl and protease inhibitors (Calbiochem). Eluates were stored at -80 °C before analysis.

The Ab concentration in blood and CVFs at each sampling was evaluated by ELISA as described above. A standard curve was created using a series of dilutions of 10-1074 IgG in PBS, ranging from 12 µg/mL to 0 µg/mL. The 10-1074 HEC gel was used at the same concentration as the 10-1074 Ab. Seven five-fold dilutions of plasma samples and CVFs (starting from 1:20 for sera, 1:100 for CVFs collected up to 6 h, and 1:20 for CVFs collected between 24 and 72 h, respectively) were added to plates in duplicate in PBS and incubated for 2 h at 37 °C. After incubation, plates were washed with PBST and incubated for 1 h with horseradish peroxidase (HRP)-conjugated goat anti-human IgG (AbSerotec) and then developed with 3,3′,5,5′-tetramethylbenzidine (TMB) substrate (Sigma-Aldrich). The development reaction was stopped with 1 M HCL and plates read at 450 nm with a plate reader (Tecan). Background values given by incubation of PBS alone in coated wells were subtracted. When assessing CVFs, results were adjusted for the dilution factor of each sample.

The Ab activity was assessed in a TZM-bl neutralization assay against SHIV$_{162P3}$. 50 tissue culture infectious dose (TCID)50 of SHIV$_{162P3}$ was incubated with a serial 4-fold dilution (starting from 1/50) of each CVL and with a serial 2-fold dilution of 10-074 IgG (ranging from 4 µg/mL to 0 µg/mL). After 1 h, thawed TMZ-bl cells ($10^4$ cells/well) were added. All conditions were done in triplicates. The plates were incubated for 48 h (37 °C, 5% $CO_2$). Subsequently, 120 µl of supernatants were removed, 75 µl of the luciferase substrate Steadylite (Perkin Elmer, Life Sciences, Zaventem, Belgium) were added to the wells, and the plates were incubated at room temperature on an orbital shaker for 10 minutes. Next, the luciferase activity was measured using a TriStar LB941 luminometer (Berthold Technologies GmbH & Co. KG., Bad Wildbad, Germany) and expressed as relative light units (RLU) and expressed as a percentage of that in positive control wells. IC$_{50}$ was calculated in GraphPad Prism 9.2.0 using non-linear regression (GraphPad Software, San Diego, CA, USA) and used to estimate the compound concentrations in CVL.

### Efficacy trial: in vivo cervicovaginal exposure to SHIV$_{162P3}$-infected splenocytes

Vaginally applied HEC gel containing 5 mg/g of 10-1074 Ab was assessed for its ability to protect against once-weekly vaginal exposure to SHIV$_{162P3}$-infected splenocytes (Fig. 3). Twelve female CMs were treated with 30 mg/kg intramuscular injection of medroxyprogesterone acetate (Depo-Provera®, Pfizer). Four weeks later, 2 mL of 10-1074 antibody loaded in HEC gel or 2 mL of placebo gel (no antibody loaded in the HEC gel) were applied into the vaginal vault. Before inoculation, the lower female reproductive tract was inspected for signs of pre-existing inflammation. After 1 h, the animals were challenged intra-vaginally with 1 mL of 10 million live SHIV$_{162P3}$-infected splenocytes. More specifically, at each challenge, $5 \times 10^6$ cells from SD03 were mixed with $5 \times 10^6$ cells from either SD01 or SD02, according to the inoculation scheme described in Table 3. A lubricated nasogastric tube (Centravet) was used to inoculate cells into the vaginal vault. After both gel inoculation and viral challenge, animals were kept in a prone position for 15 min and were monitored for product leakage.

Splenocytes were thawed 1 h before the challenge, and washed twice, then 5 million cells from each of the two donors were mixed and suspended in 1 mL RPMI medium for administration to macaques. Macaques were scheduled to receive 8 once-weekly exposures (corresponding to day 49 after the first challenge). Once animals were confirmed as virus positive by two consecutive weeks of detectable viral RNA (vRNA) in plasma (>100 viral RNA copies per milliliter of plasma), gel inoculation and viral challenge were stopped. Blood was

collected from sedated animals at baseline (before the first challenge), weekly during the challenge phase and until week 12 (day 84) after the first challenge, and then at weeks 15, 19, 23, 31, 39, 43, 52 and 58 (the latter ones corresponding to the time of euthanasia). CVFs were not collected during the challenge period to avoid possible mechanical damage to the vaginal vault and thus to facilitate the pervasion of the virus that would introduce uncontrolled changes in the value of each challenge in the repeated challenge model. 15 weeks after the first challenge, inguinal lymph node (LN) biopsies were collected to quantify viral DNA and to confirm the infectious status of the animals. Samples at euthanasia included blood, the female reproductive tract, spleen, and inguinal and iliac LNs.

### Quantification of the viral load in plasma, culture supernatants, and tissues

Blood plasma was isolated from EDTA-treated blood samples by centrifugation for 10 min at 1,500 x g and stored frozen at -80 °C. vRNA was prepared from 200 µL or 500 µL of cell-free plasma using Nucleospin 96 virus core kit (Machery-Nagel) or QIAamp ultraSens virus kit (Qiagen), respectively. Culture supernatants were collected at the indicated time points, centrifuged for 10 min at 1,500 x g, and stored frozen at -80 °C. Retro-transcription and cDNA amplification and quantification were performed in duplicate by RT-qPCR using the Superscript III Platinum one-step quantitative RT-PCR system (Invitrogen, Carlsbad, USA). RT-qPCR was performed as described in[68]. The sequence of the primer for reverse transcription was 5′-CAATTTTACCCAGGCATTTAATGTT-3′ (25 bp). PCR was carried out using the same primer and with the sense primer 5′-GCAGAGGAGGAAATTACCCAGTAC-3′ (24 bp). The TaqMan probe sequence was 5′-TGTCCACCTGCCATTAAGCCCGA-3′ (23 bp). Samples were heated for 30 min at 46 °C and 4 min at 95 °C, followed by 50 cycles of 95 °C for 15 s and 60 °C for 1 min. The lower limit of quantification (LOQ) was estimated to be 111 and 1,000 copies/mL and the lower limit of detection (LOD) was 37 and 333 copies/mL when quantifying plasma and culture supernatant respectively. The SHIV DNA copy numbers in tissues were measured by quantitative PCR, using primers amplifying the gag region of SHIV. The lower LOD was 46 copies per million cells.

### ELISA-based analyses of seroconversion to SHIV$_{162P3}$

Blood serum was isolated from a blood sample collected in a serum separation tube (SST, Fischer Scientific) by spinning for 10 minutes at 1500 g, and cryopreserved at -80 °C. ELISA assay was performed using the Genscreen HIV-1/2 Version 2 kit (BioRad Laboratories, France), according to the manufacturer's instructions. A reference HIV+ and HIV- human serum were used as positive and negative controls, respectively. The cutoff was defined as 3-fold the value of the negative control (gp160-coated wells with a SHIV$_{162P3}$-uninfected plasma sample).

### Enzyme-linked Immunospot Assay (ELISPOT) for the detection of Gag and Env-specific T cell responses

IFNγ ELISpot assay of PBMC and lymph-node mononuclear cells (LNMC) was performed using monkey IFN-γ ELISpot PRO kit (Mabtech Monkey IFN-g ELISPOT pro, #3421M-2APT) according to the manufacturer's instructions. Cells were plated at a concentration of 200,000 per well and were stimulated with gag SIV$_{mac251}$ and Env SHIV$_{162P3}$ 15-mer overlapping by 11 peptide pools at a final concentration of 2 µg/mL of each peptide. Plates were incubated for 18 h at +37 °C in an atmosphere containing 5% CO2, then washed 5 times with PBS and incubated for 2 h at +37 °C with a biotinylated anti-IFNγ antibody. After 5 washes, spots were developed by adding 0.45 µm-filtered ready-to-use BCIP/NBT-plus substrate solution and counted with an automated ELISpot reader ELR08IFL (Autoimmun Diagnostika GmbH, Strassberg, Germany). Spot forming units (SFU) per $10^6$ PBMCs are means of duplicates for each animal.

## Data visualization and statistical analysis

All data visualization and statistical analysis were performed using GraphPad Prism version 9.2.0 software (GraphPad Software, La Jolla, USA). Dose-response inhibition curves were plotted using sigmoid dose-response curves (variable slope). Wilcoxon rank-sum test was used to determine if there were any differences in weight (kg) and age (years) between the placebo controls and 10-1074 gel-treated animals at the beginning of the efficacy study. Survival analysis was used to compare time to infection in 10-1074 gel-treated animals relative to the placebo group. A Kaplan–Meier survival analysis was computed and graphed. The log-rank test was computed to determine a statistically significant difference in time to infection between the two groups of macaques. Protection efficacy was calculated using the following formula: 1- (infection rate in Ab-treated animals/infection rate in control animals) x 100. p values of 0.05 or lower were considered significant.

## Study approval

All work related to animals was conducted in compliance with institutional guidelines and protocols approved by the local ethics committee "Comité d'Ethique en Expérimentation Animale du Commissariat à l'Energie Atomique et aux Energies Alternatives" (CEtEA #44). The study was authorized by the "Research, Innovation and Education Ministry" under registration numbers APAFIS#373 2015032511332650 and #32029-2021061709451888.

## Reporting summary

Further information on research design is available in the Nature Portfolio Reporting Summary linked to this article.

## Data availability

All data are available in the main text or the supplementary materials. Source data are provided with this paper.

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

## Acknowledgements

We acknowledge B. Delache and S. Langlois for the challenge experiments; M. Leonec, L. Pintore, L. Junges, and K. Lheureux for viral load quantification; J. Morin for ELISPOT experiments analysis; L. Moenne-Loccoz for ELISA experiments; A-S. Gallouet and W. Gros for flow cytometry assistance; M. Barendji, J. Dinh, and E. Guyon for the management of NHP biological resources; F. Ducancel, Y. Gorin, B. Targat, S. Keser and I. Mangeot for their help with the logistics, safety, and resource management, and all members of LFC, ASW, LIBI, and L2I laboratories of IDMIT infrastructure for their excellent expertise and outstanding contribution. Funding. This work was funded by the French "Agence Nationale de Recherches sur le Sida et les Hépatites Virales et Maladie Infectieuses Emergentes" (ANRS-MIE, decision n°18062/19094, to MC). KS was a recipient of an ANRS-MIE doctoral fellowship. This work was also supported by the "Programme Investissements d'Avenir" (PIA) managed by the "Agence Nationale de la Recherche" (ANR) under reference ANR-11-INBS-0008, funding the Infectious Disease Models and Innovative Therapies (IDMIT, Fontenay-aux-Roses, France)

infrastructure, and ANR-10-EQPX-02-01, funding the FlowCyTech facility (IDMIT, Fontenay-aux-Roses, France). The funders had no role in the design of the study, data collection or interpretation, or the decision to submit the work for publication.

## Author contributions

Study conception and design: M.C., D.D., and R.L.G. Acquisition of the data: K.S., D.D., V.L., N.D., L.B., M.G.L., M.C. Analysis and interpretation of the data: K.S., N.D., D.D., V.L., H.M., N.D.B., M.C. HEC gel preparation: K.B. Preparation of HEC placebo gel and 10-1074 Ab: V.L., H.M. Draft of the manuscript: M.C. Preparation of figures/tables: K.S., M.C., N.D.B. Funding acquisition: M.C. and R.L.G. All authors corrected and approved the final version of the manuscript.

## Competing interests

The authors declare no competing interests.
