## [Peer Review File · Nature Communications]

Mucosal application of the broadly neutralizing antibody 10-1074 protects macaques from cell-associated SHIV vaginal exposureReviewers' Comments:

Reviewer #1:

Remarks to the Author:

Globally, most HIV infections occur because of vaginal transmission. In the absence of an effective vaccine, novel HIV prevention strategies are desperately needed. The development of monoclonal antibodies (mAbs) as an effective microbicide would be an advancement in the field to prevent infection. Animal models indicate that infected cells can efficiently transmit HIV across the vaginal and rectal mucosae and suggest that cell-associated virus may be more resistant to mAbs and ARV therapy. This manuscript discusses an experimental study conducted in cynomolgus macaques, to assess the protective potential of a broadly neutralizing HIV antibody, 10-1074, to inhibit cell-associated vaginal transmission. The antibody was formulated in gel, applied into the vaginal vault of macaques one hour prior to the vaginal exposure of SHIV162P3-infected splenocytes. This was done on weekly basis and followed for 8 wks. A key finding of this study is that mAb-treated group had significantly lower virus acquisition than the placebo group in cell-associated SHIV-vaginal challenge model, providing a proof of concept to use monoclonals as potential microbicides. The implication of this study is very important as a concept of woman-controlled method for preventing HIV acquisition by sexual contact through the development of microbicide. The manuscript is well written, and experiments are straight forward along with conventional techniques and methods used by this group are very well published in the area. The results are informative as translational application in this limited preclinical setting. This manuscript genuinely sets up a stage to consider the addition of vaginal microbicide platform to prevent cell-associated HIV transmission.

The significance of the study is unquestionable, but authors are requested to address these caveats and suggestions for improvement:

- It remains unclear how the model translates to prevention of human infection, in term of the Ab concentration required to achieve protection, concentration of Ab is missing in the treated group to better understand the correlate of protection.
- What was the criteria of using 5mg 10-1074 in the gel (any dose escalation studies were performed)?
- This manuscript could be highly significant if a cocktail of anti-HIV Env antibodies is identified to significantly inhibit cell-associated vaginal SHIV transmission. A combination of two mAbs may be sufficient to give widespread coverage against a range of clinical isolates and to avoid viral escape.
- It addresses only vaginal transmission of HIV, and an addition to a rectal model would have been a better scenario.
- What is the safety profile of gel? Any sign of irritation or inflammation observed?
- Acronym CVF needs to be described at the first place in manuscript.
- Needs more description- How well is the mAb absorbed in the vaginal tract and what is its stability in vivo?
- Monkeys are named CM01.....throughout the manuscript whereas Fig. 5 B and C mentioned different animal IDs thereby creating a confusion.
-

Reviewer #2:

Remarks to the Author:

What are the noteworthy results? Here the authors show a significant block of infection when treated with 10-1074 and using cell-associated virus (cells versus cell-free virions). The fact that 10-1074 can neutralize and block infection with SHIV162P3 is not noteworthy. Nor is the route of infection unique. But the challenge model is particularly notable. Given that there are many infected cells in semen, it is completely reasonable to test cell-associated virus in various transmission studies. However, the challenge is the implementation of such studies. It is clear that cell-free virus alone can cause infection. In contrast, the very nature of cell-to-cell infection involves both infected cell and virus budding from those infected cells. I think one way to really clarify the challenge model would be to experimentally determine how much cell-free virus is being produced over the first few hours after

thawing in culture. This would provide the reader with context about how much cell-free virus would be available in the female genital tract. In previous studies from this group, they note that there are multiple millions of viral mRNA in the cells used for infection.

Will the work be of significance to the field and related fields? I think that the research community will be happy to know that antibody interventions can be utilized in cell-associated virus (with all the caveats noted above).

How does it compare to the established literature? There are very few studies comparing cell-associated infection to cell-free. There are no papers describing protection from infection using bNAbs and cell-associated mucosal infection.

Does the work support the conclusions and claims, or is additional evidence needed? I think the best evidence that supports the conclusion that you are blocking cell-mediated infection is to validate that the model producing little to no cell-free virus during the infection event. If there are significant amounts of virus present shortly after exposure, then this must be acknowledged in title, abstract and conclusions.

Are there any flaws in the data analysis, interpretation and conclusions? - Do these prohibit publication or require revision? No.

Is the methodology sound? Does the work meet the expected standards in your field? Yes.

Is there enough detail provided in the methods for the work to be reproduced? Yes.

Reviewer #3:

Remarks to the Author:

Suphaphiphat et al. have conducted a study in cynomolgus macaques examining the efficacy of mucosal application of the broadly neutralizing antibody (bNAb) 10-1074 as a preventative treatment for SHIV exposure. The study described in the manuscript has both sound methodology that may be reproduced using the methods section provided and builds on the large breadth of data on broadly neutralizing antibodies in the context of HIV and SHIV prophylaxis. Both ex vivo and macaque model work on the subject tend to focus on blocking infection from cell-free virus, but recent studies examining ex vivo cell-cell viral transmission, including papers by Dr. Christina Ochsenbauer's group at University of Alabama, Birmingham, and a previous paper by Suphaphiphat et al. (EBioMedicine, 2020), have shown that broadly neutralizing antibodies are capable of inhibiting transmission albeit to different degrees than in the standard cell-free TZM-bl assays. Parsons et al. (Sci. Transl. Med., 2017) has previously shown that PGT121 administration conferred partial protection against cell-associated SHIVSF162-P3 transmission in macaques, but these animals were infected intravenously.

Suphaphiphat et al., conversely, have used SHIVSF162-P3-infected splenocytes in an intravaginal challenge with mucosal application of the 10-1074 antibody. As such, this manuscript contains data that is both noteworthy and of significance to the study of preventing HIV acquisition, and I believe that the manuscript should be accepted with some revisions.

My main concerns with this paper regard the high frequency of virological control in the sham-treated animals. Only two out of the five animals that became infected in that group had positive viral loads a year out from the start of the study with the other three virologically controlling. Of the treatment group, one of the six animals became infected and subsequently controlled viremia, and the authors argue that this outcome was owing to the blunted peak viremia resulting from the bNAb treatment. While I can see that the bNAb treatment could have caused the blunted peak viremia, I remain unconvinced of it necessarily owed to similar results being observed in one of the control animals as well. The ELISpot results in Figure 5 even appear to argue against it, given that the animal has cellular

IFNg responses generally in line with those of the control animals.

Additionally, I am rather surprised by the frequency of control in the cynomolgus macaques. Polacino et al. (J Med Primatol, 2008; PMID: 19187427) examined SHIVSF162-P4 infection in cynomolgus macaques and rhesus macaques. Their general observation was that cynomolgus macaques were less likely to control viremia than rhesus macaques. However, many studies from the laboratory of Dan Barouch have rhesus macaques displaying virologic control post-infection in sham treatment groups at far lower frequencies than seen in this study by Suphaphiphat et al. They mention that the high frequency of control has been seen in previous studies by the authors, so this could be an artefact of the differences in methods of infection. However, I feel this point requires slightly more discussion in the text of the paper to address any concerns about the frequency of control.

Minor Errors:

1. Line 100: "pro-viral" is used instead of "proviral".
2. Lines 104, 105, 157-9: Number formatting needs to be consistent throughout the paper, and mainland European notation is used here (ex. 40 000 instead of 40,000 and 1,8 instead of 1.8).
3. Line 117: Please write out what HEC stands for, since this is the first place the abbreviation is used.
4. Line 128, 144, 769: Depo Provera needs to be capitalized.
5. Line 129: Please write out what CVF stands for, as this is the first time the abbreviation is used in the paper.
6. Line 171: "Pick" was written at the end of the line when you meant "peak".
7. Lines 180-4: Please add at least a supplementary figure to show this data.
8. Lines 215-6: Please clarify that your previous study using semen cells was *ex vivo*, as the sentence makes it sound as if you've previously performed the major experiment with those cells, which would be a more physiological challenge than the splenocytes.
9. Figure 1: Please include X-axis labels for panels C & D.
10. Figure 2: Please standardize the animal names. PK01 & PK02 are used in the figure, but another set of names are used in the legend (Line 760).
11. Line 775: "Weekly" should be used instead of "weakly".
12. Figure 5: Animal names are not the same in panels B & C as in the rest of the figure and paper.

Sincerely,
Victoria Walker-Sperling

Point-by-point answer to the reviewer's comments.

We extend our sincere gratitude to the reviewers for their highly positive evaluation of our work and their valuable and constructive comments. We kindly request the reviewers to take into account the revised version of the manuscript (which we are sending with track changes) whenever we refer to specific line numbers in the document. Your feedback has been invaluable in improving the quality of our research, and we truly appreciate your thoughtful insights.

Sincerely,

Mariangela Cavarelli on behalf of all authors

Reviewer #1 (Remarks to the Author):

Globally, most HIV infections occur because of vaginal transmission. In the absence of an effective vaccine, novel HIV prevention strategies are desperately needed. The development of monoclonal antibodies (mAbs) as an effective microbicide would be an advancement in the field to prevent infection. Animal models indicate that infected cells can efficiently transmit HIV across the vaginal and rectal mucosae and suggest that cell-associated virus may be more resistant to mAbs and ARV therapy. This manuscript discusses an experimental study conducted in cynomolgus macaques, to assess the protective potential of a broadly neutralizing HIV antibody, 10-1074, to inhibit cell-associated vaginal transmission. The antibody was formulated in gel, applied into the vaginal vault of macaques one hour prior to the vaginal exposure of SHIV162P3-infected splenocytes. This was done on weekly basis and followed for 8 wks. A key finding of this study is that mAb-treated group had significantly lower virus acquisition than the placebo group in cell-associated SHIV-vaginal challenge model, providing a proof of concept to use monoclonals as potential microbicides. The implication of this study is very important as a concept of woman-controlled method for preventing HIV acquisition by sexual contact through the development of microbicide. The manuscript is well written, and experiments are straight forward along with conventional techniques and methods used by this group are very well published in the area. The results are informative as translational application in this limited preclinical setting. This manuscript genuinely sets up a stage to consider the addition of vaginal microbicide platform to prevent cell-associated HIV transmission.

Reply: *We are grateful to the reviewer for recognizing the merit of our study.*

The significance of the study is unquestionable, but authors are requested to address these caveats and suggestions for improvement:

- It remains unclear how the model translates to prevention of human infection, in term of the Ab concentration required to achieve protection, concentration of Ab is missing in the treated group to better understand the correlate of protection.

Reply: *We acknowledge the importance of addressing correlates of protection.*

To prevent any physical damage to the vaginal vault during the challenge period, we refrained from assessing antibody concentration in the treated group. Nonetheless, as part of the pharmacokinetics study, we did measure antibody concentration in cervicovaginal fluids (CVFs) from two animals exposed to the same antibody dose. Figure 2B-C illustrates the results, indicating a detectable mean concentration of 4 mg/ml up to 6 hours after gel application, closely aligning with the inoculated dose of 5 mg/ml. Based on these findings and considering the in vitro IC50 against cell-associated SHIV_{162P3} (refer to Table 2), we can infer that an in vivo antibody concentration above 4 mg/mL, equivalent to >10,000-fold the IC50, is crucial for achieving protection.

We have included a dedicated sentence in the discussion (lines 275-289) to elucidate the translational relevance of our results to human studies.

-What was the criteria of using 5mg 10-1074 in the gel (any dose escalation studies were performed)?

Reply: *The selection of the 5mg dose was based on several well-considered factors. Firstly, we relied on our previously published data (Moog et al., Mucosal Immunology 2012), which evaluated first-generation bNAbs 4E10, 2G12, and 2F5 against cell-free SHIV_{162P3} transmission in cynomolgus macaques. In this study, a total gel concentration of 60 mg/ml, containing 20 mg/ml of each of the 4E10, 2G12, and 2F5 NAbs, was utilized. Additionally, our findings from a separate study (Suphaphiphat et al., EBioMed 2020) demonstrated the higher inhibiting potency of 10-1074 against in vitro cell-associated SHIV_{162P3} transmission, compared to a pool of 4E10, 2G12, and 2F5Abs (mean IC90 in a PBMC assay of 1.56 ± 1.09 mg/ml versus 53.45 ± 12.65 mg/ml). These compelling data led us to reduce the dose from 20 mg to a more clinically relevant 5 mg, aiming to assess in vivo protection by 10-1074.*

Furthermore, the dose of 5 mg was rigorously tested and validated in the PK study. The results confirmed the retention of 10-1074 in CVFs at concentrations 10,000-fold higher than those required to inhibit 50% of infection in vitro during the first 6 hours. This validation provided strong support for employing the 5mg dose of 10-1074 in the subsequent protection study.

-This manuscript could be highly significant if a cocktail of anti-HIV Env antibodies is identified to significantly inhibit cell-associated vaginal SHIV transmission. A combination of two mAbs may be sufficient to give widespread coverage against a range of clinical isolates and to avoid viral escape.

Reply: *We concur with the reviewer's perspective that a combination of bNAbs is expected to confer greater protection against a variety of clinical isolates, as supported by multiple reports from other researchers and our own group. However, the primary objective of the present study was to establish the proof of concept that in vivo protection against cell-associated HIV infection can be achieved using a single bNAb.*

While we acknowledge the significance of evaluating antibody combinations in future investigations, we have already addressed this aspect in the original manuscript's discussion section (lines 298-306). The included statement emphasizes that combining bNAbs is likely to enhance the observed protective effects, capitalizing on their heightened potency and broader efficacy. Such combinations may also facilitate dose-sparing effects while guarding against the potential emergence of resistant variants.

-It addresses only vaginal transmission of HIV, and an addition to a rectal model would have been a better scenario.

Reply: *Rectal transmission undoubtedly constitutes a significant route of viral acquisition, alongside the vaginal route. In our future research endeavors, we plan to conduct a study evaluating the efficacy of Ab-based microbicides against cell-associated rectal transmission. In response to the valuable observation made by the reviewer, we have included a dedicated section in the discussion (lines 352-354) to emphasize the importance of rectal transmission and its consideration in our future investigations.*

- What is the safety profile of gel? Any sign of irritation or inflammation observed?

Reply: *We thank the reviewer for the insightful question regarding the safety profile of the gel used in our study. To avoid any potential mechanical damage of the vaginal mucosa, we*

refrained from performing colposcopy to assess local inflammation. However, it is important to note that we did not anticipate an inflammation event induced by the application of HEC gel. There are two primary reasons for this expectation. Firstly, the HEC gel formulation is widely used for vaginal delivery (Andrews, G. P. et al., Biomacromolecules 2009) and has been utilized successfully by our group in four previous protection studies (Moog C. et al, Mucosal Immunology 2014; Ariën, K. et al, Sci. Reports 2016; Le Grand R. et al, J. of Virology 2016; Dereuddre-Bosquet N. et al, Plos Pathogens 2012). Its track record of safe application and absence of any sign of irritation or inflammation in these studies provided us with confidence in its suitability for use in our current research. Additionally, it is worth mentioning that a similar HEC gel formulation has been utilized in clinical settings to assess Tenofovir safety (Abdool Karim Q. et al, Science 2010). The clinical use of the HEC gel further supports its safety profile and suggests that any potential irritation or inflammation is not a typical concern associated with its application. Nevertheless, we acknowledge the importance of safety considerations and we have included a statement in the manuscript results (lines 168-169).

- Acronym CVF needs to be described at the first place in manuscript.

Reply: *We have indicated the meaning of the CVF acronym at line 150 of the revised manuscript.*

- Needs more description- How well is the mAb absorbed in the vaginal tract and what is its stability in vivo?

Reply: *In response to the reviewer's suggestion, we have provided indication about the successful Ab absorption of the antibody in the vaginal tract (line 167) and incorporated an additional section into the discussion to emphasize its stability (lines 275-283). In vivo, the pharmacokinetic study substantiated the stability of the antibody up to 6h, as indicated by the anti-HIV neutralization activity using CVFs collected up to 72 hours (figure 2C). Similar time of retention were observed by other studies evaluating gel formulated molecules, such as neutralizing antibodies, CD4 mimetic or chemokine analogs (Veazey, R. S. et al. JID 2009; Moog C. et al, Mucosal Immunology 2014; Dereuddre-Bosquet, N. et al. Plos Pathogens 2012; Ariën, K. et al, Sci. Reports 2016). In vivo Ab stability was further confirmed by the significant protection observed in the animals challenged 1h after the application of the gel. This evidence underscores the favorable characteristics of the antibody formulation within*

the vaginal environment, in this specific setting. Whether the Ab-HEC gel will protect from a challenge at 24-72h requires further investigation.

- Monkeys are named CM01.....throughout the manuscript whereas Fig. 5 B and C mentioned different animal IDs thereby creating a confusion.

Reply: *ID codes have been corrected in Figures 5 B, C.*

Reviewer #2 (Remarks to the Author):

What are the noteworthy results? Here the authors show a significant block of infection when treated with 10-1074 and using cell-associated virus (cells versus cell-free virions). The fact that 10-1074 can neutralize and block infection with SHIV162P3 is not noteworthy. Nor is the route of infection unique. But the challenge model is particularly notable. Given that there are many infected cells in semen, it is completely reasonable to test cell-associated virus in various transmission studies. However, the challenge is the implementation of such studies. It is clear that cell-free virus alone can cause infection. In contrast, the very nature of cell-to-cell infection involves both infected cell and virus budding from those infected cells. I think one way to really clarify the challenge model would be to experimentally determine how much cell-free virus is being produced over the first few hours after thawing in culture. This would provide the reader with context about how much cell-free virus would be available in the female genital tract. In previous studies from this group, they note that there are multiple millions of viral mRNA in the cells used for infection.

Will the work be of significance to the field and related fields? I think that the research community will be happy to know that antibody interventions can be utilized in cell-associated virus (with all the caveats noted above).

How does it compare to the established literature? There are very few studies comparing cell-associated infection to cell-free. There are no papers describing protection from infection using bNAbs and cell-associated mucosal infection.

Does the work support the conclusions and claims, or is additional evidence needed? I think the best evidence that supports the conclusion that you are blocking cell-mediated infection is to validate that the model producing little to no cell-free virus during the infection event. If

there are significant amounts of virus present shortly after exposure, then this must be acknowledged in title, abstract and conclusions.

Reply: *The reviewer's comment is extremely relevant as it underlines a critical aspect of our work, which aims to demonstrate Ab-mediated inhibition of infection transmitted by infected cells. In order to address the reviewer's concern, we have performed an in vitro experiment of splenocytes culture to assess cell-free virus production in the first hours after thawing. The results are shown in the new Supplementary Table 6 and are described in the results section (lines 212-221). Infected splenocytes were thawed and mixed at a 1:1 ratio following the same protocol as the one used to prepare the cellular inoculum for the efficacy study. Cells were cultured in RPMI medium in a 24-well plate in the absence or presence of cervicovaginal fluid (CVF) to simulate the in vivo vaginal environment. Culture supernatants were collected at baseline and after 1, 2, 6, and 24 h of culture. Cell-free viral production was assessed by RT-PCR. Viral RNA was below the lowest LOD at all the time points analyzed. These results align well with the absence of free viral particles in the supernatant after the final wash of the splenocytes as reported in the original manuscript and confirm that 10-1074 Ab is mediating cell-associated infection in our experimental system.*

Are there any flaws in the data analysis, interpretation and conclusions? - Do these prohibit publication or require revision? No.

Is the methodology sound? Does the work meet the expected standards in your field? Yes.

Is there enough detail provided in the methods for the work to be reproduced? Yes.

Reply: *The positive evaluation of our work by the reviewer is greatly appreciated.*

Reviewer #3 (Remarks to the Author):

Suphaphiphat et al. have conducted a study in cynomolgus macaques examining the efficacy of mucosal application of the broadly neutralizing antibody (bNAb) 10-1074 as a preventative treatment for SHIV exposure. The study described in the manuscript has both sound methodology that may be reproduced using the methods section provided and builds on the large breadth of data on broadly neutralizing antibodies in the context of HIV and SHIV prophylaxis. Both ex vivo and macaque model work on the subject tend to focus on blocking

infection from cell-free virus, but recent studies examining ex vivo cell-cell viral transmission, including papers by Dr. Christina Ochsenbauer's group at University of Alabama, Birmingham, and a previous paper by Suphaphiphat et al. (EBioMedicine, 2020), have shown that broadly neutralizing antibodies are capable of inhibiting transmission albeit to different degrees than in the standard cell-free TZM-bl assays. Parsons et al. (Sci. Transl. Med., 2017) has previously shown that PGT121 administration conferred partial protection against cell-associated SHIVSF162-P3 transmission in macaques, but these animals were infected intravenously. Suphaphiphat et al., conversely, have used SHIVSF162-P3-infected splenocytes in an intravaginal challenge with mucosal application of the 10-1074 antibody. As such, this manuscript contains data that is both noteworthy and of significance to the study of preventing HIV acquisition, and I believe that the manuscript should be accepted with some revisions.

Reply: *We express our gratitude to the reviewer for providing a positive evaluation of our study.*

My main concerns with this paper regard the high frequency of virological control in the sham-treated animals. Only two out of the five animals that became infected in that group had positive viral loads a year out from the start of the study with the other three virologically controlling. Of the treatment group, one of the six animals became infected and subsequently controlled viremia, and the authors argue that this outcome was owing to the blunted peak viremia resulting from the bNAb treatment. While I can see that the bNAb treatment could have caused the blunted peak viremia, I remain unconvinced of it necessarily owed to similar results being observed in one of the control animals as well. The ELISpot results in Figure 5 even appear to argue against it, given that the animal has cellular IFN γ responses generally in line with those of the control animals.

Reply: *We agree with the reviewer that the Ab-treated animal that controls viremia displayed viral kinetics similar to one control macaque. We have thus deleted the sentence suggesting that the blunted peak viremia could be attributed to the Ab treatment from the abstract and the discussion (line 296) and we added a sentence at line 188 of the results to underline the similar viral kinetics observed in one control animal.*

Additionally, I am rather surprised by the frequency of control in the cynomolgus macaques. Polacino et al. (J Med Primatol, 2008; PMID: 19187427) examined SHIVSF162-P4 infection

in cynomolgus macaques and rhesus macaques. Their general observation was that cynomolgus macaques were less likely to control viremia than rhesus macaques. However, many studies from the laboratory of Dan Barouch have rhesus macaques displaying virologic control post-infection in sham treatment groups at far lower frequencies than seen in this study by Suphaphiphat et al. They mention that the high frequency of control has been seen in previous studies by the authors, so this could be an artefact of the differences in methods of infection. However, I feel this point requires slightly more discussion in the text of the paper to address any concerns about the frequency of control.

Reply: *We thank the reviewer for providing us with the opportunity to further elaborate on this aspect. We firmly believe that the observed high rate of viral control in our study can be attributed to the utilization of a low viral dose during the challenge. Additionally, we employed infected cells as the source of the virus. It is essential to note that due to the unique nature of our study, being the first in vivo investigation utilizing SHIV-infected cells, a comprehensive comparison of our results with other studies is challenging.*

Previously, our research group has garnered extensive experience in utilizing cell-free virus infection models, both with SHIV and SIV viruses. In support of our findings, we have published works showcasing rapid SHIV control following cell-free vaginal challenge (Moog C. et al, Mucosal Immunology 2014; Ariën, K. et al, Sci. Reports 2016). Moreover, we are providing confidential unpublished results herewith for the reviewer's insight, which compare low-dose intrarectal cell-free SHIV162P3 (n=14) and SIVmac251 (n=11) infections in cynomolgus macaques. Our results unequivocally demonstrate that cynomolgus macaques exhibit a higher rate of SHIV infection control compared to SIV infection.

Veazey's group reported similar results in rhesus macaques when comparing SHIV162P3 versus SIVmac infection (please see Figure 1 in Huanbin Xu et al. PlosOne 2011). In addition, rapid control of SHIV162P3 infection was observed in about half of the infected rhesus macaques by Zheng Qi et al. (Retrovirology 2012).

Notably, Polacino et al. conducted an analysis of SHIVSF162P4 infection in pigtailed macaques and rhesus macaques (not cynomolgus) and concluded that pigtail macaques are less likely to control viremia than rhesus macaques. This difference in viral kinetics between the two species is well-documented, and similar findings have been reported by other research groups (Beck S.E. et al, J. of NeuroVirology 2015). Furthermore, Polacino referred to a study by ten Haaft et a. (J Med Primatol 2001) which reported no differences in set point viral load between rhesus and cynomolgus macaques. However, this study employed SHIV89.6p, which is a highly pathogenic virus, making direct comparison with SHIV162P3 impractical. In addition, Reimann K.A., et al. (J. of Virology 2005) compared the pathogenicity of SHIV89.6p and SIVmac251 infection in cynomolgus and rhesus macaques of both Chinese origin and Indian origin. The authors concluded that by 9 to 10 months after infection, both viruses became undetectable in plasma more frequently in cynomolgus than in either Chinese or Indian rhesus macaques.

We have incorporated a dedicated discussion in the revised manuscript (lines 325-332) to address these important aspects.

Minor Errors:

1. Line 100: “pro-viral” is used instead of “proviral”.

Reply: *We corrected the typo.*

2. Lines 104, 105, 157-9: Number formatting needs to be consistent throughout the paper, and mainland European notation is used here (ex. 40 000 instead of 40,000 and 1,8 instead of 1.8).

Reply: *We changed the number formatting at lines 121-122.*

3. Line 117: Please write out what HEC stands for, since this is the first place the abbreviation is used.

Reply: *We have indicated that HEC stands for Hydroxyethylcellulose at line 134 of the revised manuscript.*

4. Line 128, 144, 769: Depo Provera needs to be capitalized.

Reply: *This has been done.*

5. Line 129: Please write out what CVF stands for, as this is the first time the abbreviation is used in the paper.

Reply: *We have indicated that CVF stands for cervicovaginal fluids at line 150 of the revised manuscript.*

6. Line 171: “Pick” was written at the end of the line when you meant “peak”.

Reply: *We corrected the error.*

7. Lines 180-4: Please add at least a supplementary figure to show this data.

Reply: *The data are shown in the new Supplementary figure 2 of the revised manuscript.*

8. Lines 215-6: Please clarify that your previous study using semen cells was ex vivo, as the sentence makes it sound as if you’ve previously performed the major experiment with those cells, which would be a more physiological challenge than the splenocytes.

Reply: *We thank the reviewer for noticing the missing information. We have now specified at line 253 of the revised manuscript that the experiments were done “ex vivo”.*

9. Figure 1: Please include X-axis labels for panels C & D.

Reply: *We have included the X-axis labels for panels C & D.*

10. Figure 2: Please standardize the animal names. PK01 & PK02 are used in the figure, but another set of names are used in the legend (Line 760).

Reply: *We corrected the names in the figure legend.*

11. Line 775: “Weekly” should be used instead of “weakly”.

Reply: *We corrected the error.*

12. Figure 5: Animal names are not the same in panels B & C as in the rest of the figure and paper.

Reply: *Animal names have been changed in figure 5, panels A and B.*

Reviewers' Comments:

Reviewer #1:

Remarks to the Author:

All concerns have been addressed in the revision.

Reviewer #2:

Remarks to the Author:

Glad that experiment worked well to answer the question of virus produced during infection.

Reviewer #3:

Remarks to the Author:

Regarding the study by Suphaphiphat et al. in cynomolgus macaques examining the efficacy of mucosal application of the broadly neutralizing antibody (bNAb) 10-1074 as a preventative treatment for SHIV exposure, the manuscript continues to have sound methodology that may be reproduced using the methods section provided, build on the large breadth of data on broadly neutralizing antibodies in the context of HIV and SHIV prophylaxis, and contain data that is both noteworthy and of significance to the study of preventing HIV acquisition.

Both the responses the authors have given to myself and the other reviewers and the concomitant changes to the manuscript are thorough and convincing in assuaging any previous concerns I had regarding the paper's publication. As such, I now recommend the paper to be accepted with no further revisions.

Sincerely,

Victoria Walker-Sperling